# Classification of Bladder Emptying Patterns by LSTM Neural Network Trained Using Acoustic Signatures

**DOI:** 10.3390/s21165328

**Published:** 2021-08-06

**Authors:** Jie Jin, Youngbeen Chung, Wanseung Kim, Yonggi Heo, Jinyong Jeon, Jeongkyu Hoh, Junhong Park, Jungki Jo

**Affiliations:** 1School of Electromechanical and Automotive Engineering, Yantai University, 30 Qingquan Road, Laishan District, Yantai 264005, China; jinjie910@sina.com; 2Department of Mechanical Engineering, Hanyang University, Wangsimni-ro 222, Seongdong-Gu, Seoul 04763, Korea; chung_911@naver.com (Y.C.); xqxwxexr@naver.com (W.K.); 3Department of Medical and Digital Engineering, Hanyang University, Wangsimni-ro 222, Seongdong-Gu, Seoul 04763, Korea; dydrlgj@gmail.com; 4Department of Architectural Engineering, Hanyang University, Wangsimni-ro 222, Seongdong-Gu, Seoul 04763, Korea; jyjeon@hanyang.ac.kr; 5Department of Obstetrics and Gynecology, Hanyang University Seoul Hospital, Wangsimni-ro 222, Seongdong-Gu, Seoul 04763, Korea; hohjk@hanyang.ac.kr; 6Department of Urology, Hanyang University Seoul Hospital, Wangsimni-ro 222, Seongdong-Gu, Seoul 04763, Korea

**Keywords:** acoustic signal, classification, flowrate prediction, lower urinary tract symptoms, long short-term memory

## Abstract

(1) Background: Non-invasive uroflowmetry is used in clinical practice for diagnosing lower urinary tract symptoms (LUTS) and the health status of a patient. To establish a smart system for measuring the flowrate during urination without any temporospatial constraints for patients with a urinary disorder, the acoustic signatures from the uroflow of patients being treated for LUTS at a tertiary hospital were utilized. (2) Methods: Uroflowmetry data were collected for construction and verification of a long short-term memory (LSTM) deep-learning algorithm. The initial sample size comprised 34 patients; 27 patients were included in the final analysis. Uroflow sounds generated from flow impacts on a structure were analyzed by loudness and roughness parameters. (3) Results: A similar signal pattern to the clinical urological measurements was observed and applied for health diagnosis. (4) Conclusions: Consistent flowrate values were obtained by applying the uroflow sound samples from the randomly selected patients to the constructed model for validation. The flowrate predicted using the acoustic signature accurately demonstrated actual physical characteristics. This could be used for developing a new smart flowmetry device applicable in everyday life with minimal constraints from settings and enable remote diagnosis of urinary system diseases by objective continuous measurements of bladder emptying function.

## 1. Introduction

Urinary impairment, which causes a decline in the quality of life, increases with age. For diagnoses of the cause of LUTS, conventional weight transducer sensors and ultrasonography, as forms of uroflowmetry, have been the common non-invasive techniques for measuring bladder emptying [1,2,3]. Uroflowmetry is more objective than an IPSS or OABSS questionnaire, and they are typically performed on an outpatient basis at a specific time and place. Nonetheless, such tests have many limitations for measuring urinary patterns [4]. For example, some patients may be forced to urinate even though they have no urge, while some patients may have difficulty in urinating at a specific place, such as hospitals, in a similar manner to patients with white-coat syndrome.

Urination is influenced by the environment and emotions. Such a phenomenon is even more prominent in patients with an overactive bladder [5]. Because of these urinary characteristics, we cannot use flowmetry alone to obtain accurate urinary patterns. Porru et al. [6] applied home- and office-based uroflowmetry tests for assessing treatment outcomes in LUTS patients. They provided information that was beneficial for examining the status of LUTS through a physician’s measurement of multiple flow curves overnight in a clinical trial. Krhut et al. [4] compared the difference between sonoflow and uroflowmetry. Kwon et al. [7] proposed a test method based on the flowrate measurement using a mobile phone application. 

Diagnosis of the bladder emptying pattern needs to have information about the flowrate and total flow volume. For outside flows, it is not straightforward to install instruments [8]. The sounds and vibrations due to urine flows include important information about the health status. Proper daily collection of the statistical features helps to diagnose the status and progress. A long short-term memory (LSTM) network is a type of recurrent neural network capable of learning order dependence in sequence prediction problems [9]. The LSTM network was proposed to address the vanishing gradient problem [10]. LSTM learns to bridge minimal time lags in excess of 1000 discrete time steps through “constant error carousels” (CECs) within special units, called cells. It has advantages for predicting patterns in urinary activities due to LUTS with a long time period.

In this study, the quantification was preformatted using the sounds of urine flow with the statistical features trained by deep learning. The proposed model substitutes the classic uroflowmetry to be used in daily life. As an example, for application of the proposed methodology by the acoustic signals, the long short-term memory (LSTM) was used to classify LUTS, compared with those treated, by the information from the uroflowmetry. A dataset was constructed with loudness and roughness as the input coefficients. The flowrate measured using uroflowmetry was used as the output coefficient. The algorithm was constructed to classify the health status of the urinary system based on the prediction results of the flowrate sequence using the LSTM classification model. This allows applications to continuously monitor urination characteristics on an everyday basis.

## 2. A New Non-Invasive Method for Uroflow Measurement

When detailed information about the urination characteristics is required, the patient begins urinating into the uroflowmeter connected to a computer recorder for measuring the voided volume, voiding time, and urinary pattern. The test results assist the physician in identifying the causes of specific urological impairment and determining the efficiency of the bladder and the sphincter. In this study, the sounds from the interaction of uroflow with the test setup was utilized as a new non-invasive method. Figure 1 shows the experimental setup for measuring the uroflow sounds. To reduce the load measurement error from the microphone weight and to prevent sensor failure due to bouncing urine during urination, a microphone (PCB, PIEZOTRONICS, 132249) was set up about 30 cm from the uroflowmeter. The sound was recorded by a multi-channel device (HEAD acoustics, SQobold, 33020162). When the patient began urination, the sound pressures were collected in real time. The urine weight was recorded simultaneously using the load cell transducer. The volume flowrate, *V*, was calculated by differentiation of the urine weight as:(1)$$V\left(t\right)=\dot{W}\left(t\right)/\rho$$
where *W* and *ρ* are the urine weight and the density of urine, respectively.

## 3. Uroflow Identification with Sound Radiations

To examine the real-time fluctuations in the sound pressure according to the urine velocity, short-time Fourier transform (STFT) was performed to study the variation in spectral components depending on the flowrates as: (2)$$X\left(k,m\right)=\sum_{n=-\infty }^{\infty }p\left(n\right)W\left(n-m\right)e^{-j\left(2\pi k/N\right)n}$$
where *p* and *N* are the discrete sound pressure to be transformed and the number of periodic frequency samples, respectively (herein *N* = 4096). This transformation allowed an investigation of the influence of background noise on the experimental environment. The Hanning window was applied for leakage reduction as:(3)$$W\left(n\right)=0.5\left[1-\cos\left(\frac{2\pi n}{m}\right)\right],\;n=0,1,\ldots ,m-1$$
where *m* is the shifting length of the window function. The transformation was performed with a 25% overlap and *m* = 4096. Figure 2 shows the waveform of the measured urinary sound and STFT components. The effect of background noise occurs at a frequency range below 100 Hz, and the maximum spectrum level was 50 dB.

With the increase in outflow velocity, the uroflow sound generated from flow impact increased over the entire frequency range correspondingly. With the constant dimension of the patient’s urinary track, a rapid urinary speed generated uroflow sound with high energy at high frequencies. To enhance the accuracy of the LSTM predictions, a dataset was constructed with loudness and roughness as the input coefficients, and the flowrate measured using uroflowmetry as the output coefficient. The loudness represented the volume of acoustic signals measured by a microphone. Roughness reflected the pattern of variation in the spectrum level. These two parameters were used as the input features of LSTM for the training process. Figure 3 shows the overall flowchart for this study. Part of the dataset was used to construct the LSTM deep-learning algorithm for prediction of the flowrate sequence, while the remaining data were used to confirm the accuracy of the training model and optimize the deep-learning parameters. The algorithm was constructed to classify the health status of the urinary system based on the prediction results of the flowrate sequence using the LSTM classification model. Compared with the conventional method using a load cell and a specific program in a given simple toilet, real-time velocity measurements with the LSTM method were proposed in this study. As it is measured only by a sound sensor, it can be easily and simply applied in various environments such as public toilets and homes, as well as hospitals, for developing an automatic classification system for urinary patients.

## 4. Patients and Inclusion Criteria

This study included patients being treated for LUTS at the Department of Urology, Hanyang University Medical Center, who were admitted between April and May 2019. All study participants provided informed consent, and the study design was approved by the appropriate ethics review board. The patients’ information is shown in Table 1. Exclusion criteria were as follows: patients less than 18 years, diaper-voiding patients, or patients on self-catheterization because of a neurogenic bladder, and patients diagnosed with vesicoureteral reflux during voiding. The protocol was performed in accordance with the relevant guidelines and regulations.

The urinary flowrate pattern calculated by Equation 1 was used to divide the patients into 3 groups as shown in Figure 4, based on doctors’ clinical experience. In clinical settings, doctors classify patients’ health conditions using the maximum rate, urination time, average rate, and the pattern of the graphs. In the healthy group, the maximum flowrate was larger than 15 mL/s. The flowrate showed a bell-shaped variation, with a rapid increase in the initial phase, followed by a gradual decrease, as shown in Figure 4a. This demonstrated the beginning and the end of the external sphincter activity during urination, and voiding time was relatively short. Figure 4b shows a maximum flowrate less than 10 mL/s, where the bell-shaped pattern was not maintained. There was intermittent interruption or a decreased flowrate during urination. In such cases, lower urinary tract obstruction or an impaired detrusor contractility was suspected. The third group had a maximum flowrate of less than 5 mL/s, disrupted pattern, a delay in voiding time, and an interrupted shape depicted by examination of the reference, as shown in Figure 4c. In such cases, there was a high likelihood of urethral obstruction (stenosis, compression by tumor, or prostatic hyperplasia) or urinary impairment. Such pattern variations were used to classify and label the health status of the participants. 

Because of the limitation of the quantity of patient samples, an additional dataset was required to establish an LSTM classification model. The interpolation method and random signals were applied to obtain an additional flowrate dataset as:(4)$$V^{\prime }(t)=\left|V\left(t\right)+A\cdot rand\left(t\right)\right|,\;0<A<0.05\cdot \max\left[V\left(t\right)\right]$$
where *V*, *A*, and rand were the original flowrate measured by uroflowmetry, the amplitude, and the random signal, respectively. One hundred flowrate data (*V′*) per patient unit were additionally constructed for training the LSTM classification model; this was verified with the raw flowrate data (*V*) measured by uroflowmetry. The classification results of the patients’ health status with the LSTM model were compared with the doctor’s diagnosis.

## 5. Statistical Feature Extractions from Sounds

For continuous real-time monitoring, the measured sounds need to be converted into features representing the flow mass. The flow mass was recorded in regular time intervals during urination. For conversion of the sound response into the flow mass, the statistical features representing level variations were used. The loudness represents the sound volume. The roughness represents sound fluctuations [11]. These two parameters were calculated by a Head Artemis Analyzer and used as the input coefficients for the LSTM model. A loudness value (*N*) was calculated as the integral of the specific loudness (*N**’*) at a critical band rate as [11]:(5)$$N=\int_{0}^{24Bark}N^{\prime }dz$$

The specific loudness was calculated quantitatively as:(6)$$N^{\prime }=0.08{\left(\frac{E_{TQ}}{E_{0}}\right)}^{0.23}\left[{\left(0.5+0.5\frac{E}{E_{TQ}}\right)}^{0.23}-1\right]\frac{\mathrm{soneG}}{\mathrm{Bark}}$$
where *E_TQ_* and *E*_0_ are the excitation versus the critical band rate at a threshold of quiet and that corresponds to the reference intensity *I*_0_ = 10^−12^ W/m^2^, respectively. The index *G* at the unit “sone” was added to represent the calculation using the critical band levels. Transformation of the sound spectrum into an excitation pattern was described by [12,13]. The Hanning window was used for transformation with *N* = 4096 and a 25% overlap. 

When the sound is modulated at a frequency between 20 and 300 Hz, the change in sound according to the time cannot be felt; only the roughness of the sound is felt. Using the boundary condition of a 1-kHz tone at 60 dB and 100%, amplitude-modulated 70 Hz produces a roughness of 1 asper; the rate of the change in the sound level was evaluated by the roughness, calculated as:(7)$$R=0.3\int_{0}^{24Bark}f_{\mathrm{mod}}\Delta L\left(z\right)dz$$
where *f*_mod_ and Δ*L* represent the modulation frequency and the masking depth, respectively [11]. The modulation frequency was set to 70 Hz. With the sound pressure of the measuring environment shown in Figure 2, the masking depth was set as 50 dB to minimize the effect of the background noise on the roughness calculation. The calculated real-time sound parameters reflected the measured flowrate very closely, as shown in Figure 5. Datasets were constructed using the sound parameters by labeling them with the measured flowrate. This allowed robust estimation with minimal dependence on the surrounding environment.

## 6. LSTM Network for the LUTS Health Monitoring

LSTM is based on the architecture of an upgraded recurrent neural network (RNN) that includes the preservation of previous information. The cell state in LSTM only needs linear summation to pass through the hidden layer, and the gradient moves between networks without attenuation. LSTM makes the neural network switch between memorizing the latest information and the information from a long time ago, so that the data can decide which information to keep and which to forget. This makes it suitable for classifying, processing, and predicting time series data [10]. Figure 6 shows the LSTM network for the prediction of urinary flowrate using acoustic parameters as the input sequences. It consists of multiple cells, and each LSTM unit cell has an input gate (*i_t_*), an output gate (*o_t_*), a forget gate (*f_t_*), and a memory cell state (*C_t_*). The cells remember the values over an arbitrary time interval, while the three gates control the flow of information into and out of the cells [8]. When the input vector constructed with the uroflow sound loudness and roughness [*x_t_* = (*N_t_*, *R_t_*)] and the previous output values (*h_t_*_−1_) are given, the first step in the forget gate layer of the LSTM cell is to determine which information is to be thrown away, which is carried out as follows:(8)$$f_{t}=\sigma \left(W_{f}\left[h_{t-1},x_{t}\right]+b_{f}\right)$$

The next step is to decide what information is to be saved in the input gate layer. A vector of new candidate values is added to the state as follows:(9)$$\begin{array}{c} i_{t}=\sigma \left(W_{i}\left[h_{t-1},x_{t}\right]+b_{i}\right),\\ \tilde{C_{t}}=\tanh h\left(W_{c}\left[h_{t-1},x_{t}\right]+b_{c}\right), \end{array}$$

The output value from the forget gate (*f_t_*) is multiplied to forget a certain amount of the cell state values (*c_t_*_−1_), whereas the input (*x_t_*) and previous output values (*h_t_*_−1_) are multiplied by the processed output values of the input gate (*i_t_*) to accept a certain amount as input values, from which a new cell state (*c_t_*) is created. The output value (*o_t_*) under this cell state is multiplied as the output of the LSTM cell (*h_t_*) to determine how much to forget from the cell state value and how much of the input value should be newly accepted, as follows:(10)$$\begin{array}{c} c_{t}=f_{t}c_{t-1}+i_{t}\tilde{C_{t}},\\ o_{t}=\sigma \left(W_{o}\left[h_{t-1},x_{t}\right]+b_{o}\right), \end{array}$$

The present study proposed an LSTM for real-time flowrate estimation of the external flow and carried out training of the LSTM with the loudness and roughness of the uroflow sounds as the inputs, and the mass flowrate measured by the weight balance as the output (*V_t_*).
(11)$$V_{t}=o_{t}\tan h\left(c_{t}\right)$$

To check the health status of the urinary system in a comfortable environment, the health status classification utilizing the flowrate was further proposed. Figure 6 shows the structure of the LSTM neural network for the classification of health status after the flowrate predictions. Unlike the sequence-to-sequence LSTM prediction model, the LSTM classification model is a sequence-to-label data classification network [14]. When the predicted flowrate sequence is expressed as a vector using the look-up layer, it is applied to the LSTM network. The output (*h_t_*) from the last instance expresses the entire sequence. The softmax non-linear layer predicts the probability distribution of the fully connected layer and health status as:(12)$$y=\mathrm{softmax}\left(W_{ht}+b\right)$$
where *y*, *W*, *σ*, tanh, and *b* represent the probability of health status classification, the weight, the sigmoid function, the hyperbolic tangent, and the bias term, respectively.

## 7. Results

The input node, batch size, hidden unit parameters were set to one, three, and five, respectively. The optimizer function was set as Adam. Epoch was set to 40 because any change above this did not affect the model’s performance. To make the output layer predict only the flowrate, a dense layer with just one neuron was used to carry out the training of the LSTM network. To verify the accuracy of the flowrate predictions through the constructed LSTM, Figure 7 compares the transient variations. This comparison allows pattern recognition performance evaluation, regardless of the magnitude of the flowrate. These graphs show the comparison between the uroflowmetry measurements and the predicted flowrate by application of the loudness and roughness of uroflow sounds that were not used in the training. The results confirm that the derivative varied very closely. The constructed LSTM model allowed an estimation of health status by the variation patterns.

Figure 8 compares the flowrate of the proposed LSTM predictions with the experimental measurements. There were many matching inflection locations between the predicted and measured values, and the mean flowrate error rates were 5.9, 6.0, and 6.2%. This confirms the potential for the development and realization of an LSTM deep-learning-based uroflowmetry system that is robust against the surrounding environmental noise. For accurate training and discrimination, the existing flowrate data need to be normalized between zero and one on the time scale. The interpolation method was applied to resample the data length to 246 for constructing the training database. When the batch size, epoch, hidden unit, and class number were set to 3, 500, 50, and 3, respectively. The data obtained using the interpolation method were used for training, and accuracy was tested using the measured flowrate. Group A represented normal urinary activity with a bell-shaped urinary pattern, Group B represented LUTS or IDC with a staccato pattern, and Group C represented BPH/urethral stenosis or compression by adenoma with an interrupted pattern. The rate values predicted through the constructed classification model differed from each other. Rate values were classified according to the graph pattern, and a rate value of 0.9 or higher was derived from the specific rate. Depending on the amount of data used for training and the training parameters, the rate value may vary. In this study, the diagnosis and treatment by the medical team from the urination characteristics of patients were similar to the health status classification results found by flowrate patterns through the proposed LSTM classification model, as shown in Table 2. The technology proposed in this study can be used for predicting the flowrate based on measurement of uroflow sounds. It can also be used to check health status through the predicted flowrate patterns and enable suspicion in addition to early diagnosis of LUTS. The magnitude of the sound varies depending on the environment in which it is measured or on the sensitivity of the microphone device. Compared with the two-dimensional input using the loudness and roughness of the urinary sound, the error rate of the flowrate predicted only with roughness increased slightly, but the patterns were similarly predictable. For LSTM health status classification, the flowrate was normalized from 0 to 1, so it depended only on the shape of the pattern rather than the flow mass. Thus, the health status classification was possible only from variation in the pattern of the uroflow sound.

## 8. Discussion

As a data-driven estimation method, LSTM has feedback connections and it processes not only a single data point but also entire sequences of data. This study developed a new LSTM estimation methodology using uroflow sound signals as a replacement for conventional flowmetry. The proposed LSTM model allowed physical flowrate measurements with a high level of accuracy. The diagnosis of urinary impairment using only the sound pressure is not a novel idea. In this study, acoustic responses generated during urination in arbitrary environment were used for an LSTM algorithm to help the accurate diagnosis of urinary impairment. Many urologists have taken interest in and conducted various studies on the diagnosis of urinary impairment using sounds. Koiso et al. [15] attached a sensor to a perineum for measurement and analysis of the acoustics of urine passing through the urethra. Their results showed the ability to objectively identify bladder outlet obstruction. The basic concept of utilizing sound analysis for estimating urinary flow parameters has been verified, and conventional sono-uroflowmetry has been applied to cell phones for use in various mobile network environments [4,7]. They created a flowmeter using audio recordings of flow sounds using a smartphone application. These sono-uroflowmetry recordings were visualized in the form of a trace, representing sound level over time.

The sound level is a factor expressing the loudness of the sound. Differences may occur depending on the height of the urine fluid. The present study showed its applicability in several forms such as panels in public smart toilets. The present study used time-dependent variations in the loudness and roughness of the acoustic response generated during urination as input values. The LSTM predictions closely followed the measurements of conventional flowmetry and allowed accurate prediction of the urination patterns. By utilizing the pattern variations, the presented identification was robust to differences in the height of the urinary tract. This study classified the patients into groups from the predicted flowrate patterns based on uroflow sound. Group A represented normal urinary activity with a bell-shaped urinary pattern, Group B represented LUTS or IDC with a staccato pattern, and Group C represented BPH/urethral stenosis or compression by adenoma with an interrupted pattern. These groupings allow the monitoring of health status through the flowrate patterns, and enable suspicion in addition to early diagnosis of LUTS. This procedure would be useful in clinical practices, homes, and offices. Our algorithm is applicable to portable devices for use in various environments with minimal influence from the surrounding noise.

As the clinical data were measured on patients suffering from LTUS, the data used were collected almost exclusively from elderly males (two females and 25 males) in this study. The focus was on securing the possibility of classification techniques for LUTS patient states using the deep-learning algorithm from the engineering perspective. For mass application in the other fields, a larger number of samples in different circumstances (US or European toilets, sitting/standing, background noise, females, different age groups) should be secured. The multipath channel effects need to be overcome to reduce the signal distortion due to reflections and reverberations. Further statistical analysis needs to be conducted in the future for more robust and precise diagnoses of LUTS in various environments.

## 9. Conclusions

Repeated flowmetry at home or office with the user in a stable condition and a comfortable environment is relevant to the accurate measurement of urinary patterns. Such potential can be recognized despite the white coat syndrome often noticed during blood pressure measurements [16]. The measurements obtained by the LSTM deep-learning algorithm would be beneficial for minimizing the influence of surrounding noise. This is important because many of the patients are elderly; hence, minimizing their environmental constraints enhances the accuracy of tests.

Devices for home-based uroflowmetry have been developed in various forms [17,18,19]. The findings from this study may be used to promote the development of various devices in different formats. The disposable home device proposed by Heesakkers et al. [20] was used to measure urinary symptoms in male patients with LUTS. They developed and analyzed the device by focusing on *Q*_max_ and reported the findings from the analysis after using the device on 59 male patients. This study realized that 81% of the patients preferred measurement at home. Despite the advantages of measurement at home, there are limitations in using the device on females or children. The device developed in the present study allowed accurate measurements in both males and females.

The LSTM flowrate classification model proposed in the present study illustrated accurate diagnosis for LUTS and early diagnosis of abnormal symptoms. By integrating linkage and wireless communication technologies for IoT-based metrological, analytical, and monitoring devices, a fun device for male and female public restrooms could be developed to realize LUTS health diagnoses of abnormal signs for the public, while also making a significant contribution to accumulating national health data, which is our next activity for further research.

## Figures and Tables

**Figure 1 sensors-21-05328-f001:**
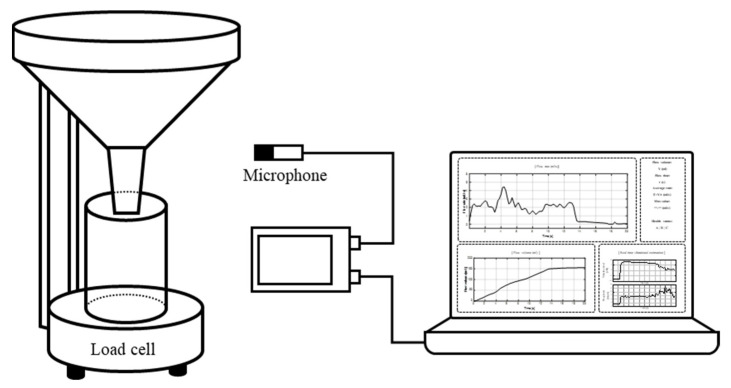
Experimental setup to measure the radiated urinary sound and weight of urine. A microphone was installed 30 cm away from the experimental setup to collect sound radiations. Uroflowmetry was also used to characterize the actual flow pattern. This pattern was utilized for LSTM training and verification.

**Figure 2 sensors-21-05328-f002:**
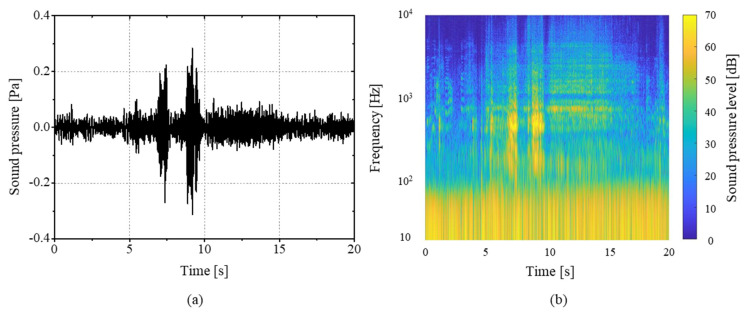
(**a**) Measured sound pressure waveform of the urine stream and (**b**) its STFT components.

**Figure 3 sensors-21-05328-f003:**
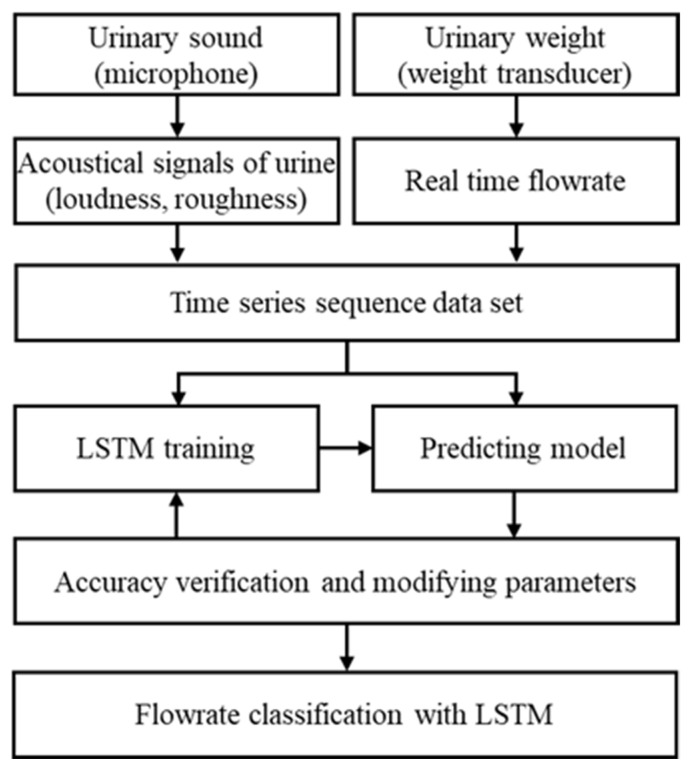
Framework of the proposed LSTM neural network for predicting and classifying the urinary flowrate with the urinary sound quality factors.

**Figure 4 sensors-21-05328-f004:**
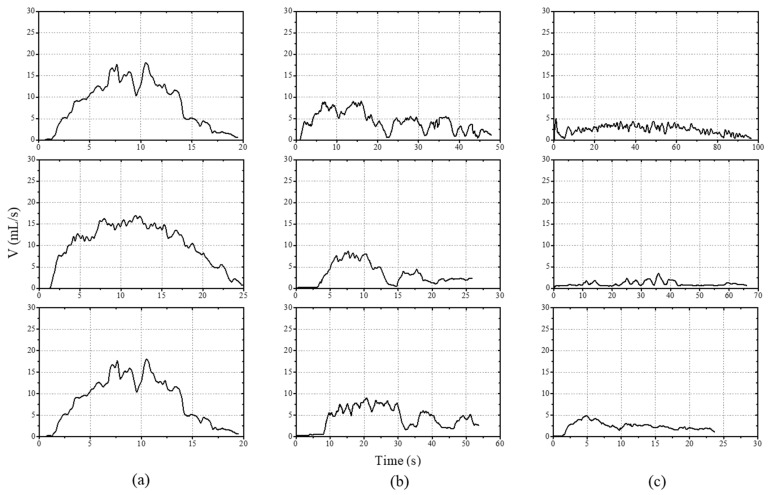
Classification of the patients’ urinary flowrate based on doctors’ clinical experience through uroflowmetry results. The measured flowrate divided the patients into three groups where (**a**) a bell-shaped variation represented the healthy patient group, (**b**) the staccato and (**c**) the interrupted flowrate indicated lower urinary track obstruction or urinary impairment.

**Figure 5 sensors-21-05328-f005:**
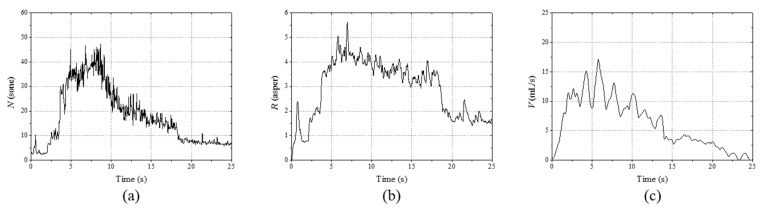
Comparison of the sound parameters and urinary information: (**a**) loudness, (**b**) roughness, and (**c**) flowrate. The time-dependent variations in the sound parameters closely followed the flowrate variations.

**Figure 6 sensors-21-05328-f006:**
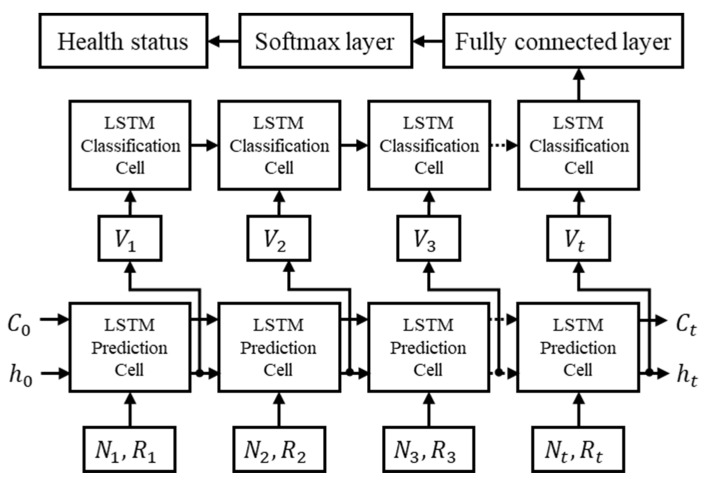
LSTM neural network for predicting a time series of urinary flowrate using sound quality factors (loudness and roughness) as input sequences and for health status classification with a curve pattern.

**Figure 7 sensors-21-05328-f007:**
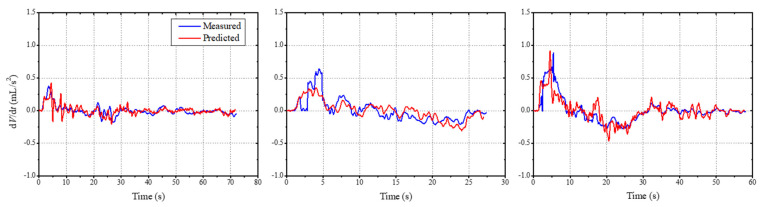
Comparison of the derivative of the predicted flowrate curves with the actual uroflow measurements.

**Figure 8 sensors-21-05328-f008:**
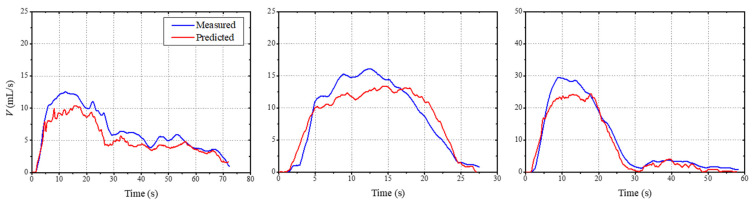
Comparison of the predicted flowrate results using the proposed LSTM model and the experimental measurements. There were many matching inflection points between the predicted and measured values, and the mean flowrate error rates were 5.9, 6.0, and 6.2%, respectively.

**Table 1 sensors-21-05328-t001:** List of the patients participating in the clinical trial after IRB approval.

Age	51–60	51–70	≥71
	5	10	12
Type	A	B	C
	15	6	6

**Table 2 sensors-21-05328-t002:** Health status classification of the patients by the LSTM algorithm.

Gender	Age	Voiding Volume	Voiding Time	Maximum Flowrate	Average Flowrate	Classification Rate			Doctor’s Diagnosis
						A	B	C	
M	70	119.9	13.3	14.8	8.6	0.9990	0.0010	0.0000	Normal
M	58	340.3	42.3	20.8	8.5	0.9979	0.0021	0.0000	
F	61	99.1	3.5	44.4	20.2	0.9995	0.0005	0.0000	
M	69	275.2	26	17	10.5	0.9992	0.0008	0.0000	
M	66	172	16.8	18.1	9.3	0.9994	0.0006	0.0000	
M	58	555	54	32.6	10.8	0.9996	0.0004	0.0000	
M	89	50.5	19	4.7	2.9	0.0000	0.9983	0.0017	LUTS or IDC
M	77	187.8	42.5	9	4.8	0.0001	0.9999	0.0001	
M	66	63.9	9.3	9.2	5.9	0.0008	0.9991	0.0001	
M	81	43.9	14	5.8	3.1	0.0001	0.9998	0.0001	
M	68	195.2	44.8	9.1	4.5	0.0000	0.9998	0.0002	
M	79	320	64.8	13.9	5.2	0.0002	0.9998	0.0000	
M	61	495.3	71	13.6	6.8	0.0002	0.9997	0.0001	
M	72	110.6	19	9.4	5.3	0.0000	0.9998	0.0002	
M	71	99.1	50.8	6.8	2.8	0.0001	0.9993	0.0006	
M	69	185	73.3	12.8	4.2	0.0000	0.9998	0.0002	
M	71	123.9	36.5	7.4	3.5	0.0001	0.9995	0.0004	
M	63	281.9	47.8	17.7	7.7	0.0005	0.9995	0.0000	
M	81	76.3	20.3	8.7	3.9	0.0000	0.9999	0.0001	
M	60	171.5	28.3	17.1	6.4	0.0001	0.9998	0.0001	
M	74	172.3	49.8	8.6	3.6	0.0002	0.9997	0.0001	
M	69	106.5	34.5	8.7	4	0.0000	0.0004	0.9996	BPH or urethral stenosis
M	51	71.9	49.8	7.4	2.7	0.0000	0.0002	0.9998	
M	85	33.7	60.8	3.5	1.9	0.0000	0.0000	1.0000	
M	83	50.3	19.8	4.9	2.2	0.0000	0.0003	0.9997	
M	75	223.5	93	5	2.6	0.0000	0.0004	0.9996	
F	72	266	61	8.3	4.3	0.0000	0.0043	0.9957	

## Data Availability

Not Applicable.

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
