# Peer review of "Classification of Bladder Emptying Patterns by LSTM Neural Network Trained Using Acoustic Signatures"

_sensors, 2021, doi:10.3390/s21165328_

Round 1

Reviewer 1 Report

The paper presents an interesting application  of LSTM to time series analysis of audio data in medical applications. I am happy to recommend it for publication. The only point that I think the  author  shall improve in a final submission is the following: The study relies on a very small sample in the clinical trial and  a significance statistical analysis is necessary to better support their findings. 

Author Response

Thank you for inviting us to submit a major revised draft of our manuscript. We appreciate the time and effort by the editor and the reviewers to provide insightful feedback for clear presentation of our work. We have incorporated changes in the manuscript that reflect the detailed suggestions of reviewers have provided. We hope that our edits and the responses provided below are satisfactorily address all the issues and concerns the reviewers have noted. To facilitate your review of our revisions, the following sections show point-by-point responses to the questions and comments.

Reviewer 2 Report

The authors present a long short-term memory recurrent neural network model in which recorded sounds of urine flow were used to predict actual urine flowrate pattern and flow parameters as measured by conventional uroflowmetry. This is an exciting research and a promising model because it is technically reliable for a non-invasive, non-contact, and affordable home uroflowmetry providing real-life data. Instead of using only sound loudness, the authors added roughness, a psychoacoustic quality of the sound, as an input signal to more accurately predict flow patterns. Presented results show low mean error rates in the test sample when comparing predicted flow pattern to the actual ones, captured by the weight-based device. Also, the classification of the flow patterns by the LSTM model was consistent with the ones done by the experts (although using an objective measure such as flow index would be preferable).

Although this research is exciting there are some reservations regarding the way how results are presented that unfortunately limits the enthusiasm.

While the authors do highlight the potential advantages of their study, there is no acknowledgment or discussion of the limitations of this study:
Dataset used were almost exclusively from elderly males (2 females and 25 males), and it may be more generalized towards elderly males, meaning it will have less accurate results in females or children (further studies in these groups is warranted). Maybe the results of the model will be as accurate as in this report but validation on the larger sample in different circumstances (US or European toilets, sitting/standing, background noise, females, different age groups) is necessary. The authors should propose further research in this matter.

It is noteworthy that this model also shares all the limitations of the uroflow test, and it only contributes to the medical history and other noninvasive and invasive procedures in urology. It can rarely be used solely in decision making. I would advise authors to avoid engaging into discussion on the diagnostic strength of the model; the fact that it can classify the patterns to the proposed types is quite enough.

Also, it is not explained how sound roughness eliminates the effect of background noise. Did the authors perform additional tests not shown to prove this?

Data preprocessing is not described in the methodology. Programming environment, and libraries used are not listed. References on equations and psychoacoustic sound quality (roughness) are not provided. The code and dataset are not uploaded to the public repositories, but if the authors prefer to keep it proprietary, it is advisory to provide a web or mobile application to give the opportunity for other researchers to validate the model in different circumstances and on different datasets. If application is not planned, authors should work on providing more pseudocode to make this work reproducible. 

Some parts of the manuscript should be corrected or explained, such as:

  1. I cannot comprehend how formula 1 and figure 2 are related to the model, there seems not to be any connection with the further computing. What is the reason for STFT? Furthermore, p and N are the number of periodic finite time and frequency sample, respectively. It is not explained how "p" is calculated from data and what was the value of N. It is not stated what was the value for shifting amount for Hanning window (also equation for Hanning function should be stated).
  2. Figure 2 legend is copied form Figure 1.
  3. Row 152-154: "For continuous monitoring and recording, the measured sounds need to be converted to features representing the flow mass. The sounds are measured in the very high
    frequency range." It is not specified what frequency spectrum of "very high frequency range" is, it is too vague.
  4. From equation 2 it is not clear how loudness was calculated because function N' is not stated. It is not clear how it is derived from row acoustic signal.
  5. After equation 3, authors should state what was the value of chosen fmod, write deltaL(z) function and how it is determined from the data, there is no clue about this.
  6. reference number 8 needs to be corrected.

Author Response

(The authors gave the same response as above.)

Reviewer 3 Report

This work uses urination acoustic signals to determine the uroflow rates and further apply the results for lower urinary tract symptom patient classifications. This approach may provide measurement devices with lighter weight and reduced size compared with existing weight differential based devices. The authors also projected the possibility of using mobile microphones at any environment setting to obtain such measurement results.

There are some fundamental issues the authors didn’t address or didn’t explain clearly:

1. The manuscript described the microphone as being at 30cm away from the flow mechanical setup to collect sound. It also claimed the measurement has minimal dependence on the surrounding environment. It didn’t describe how it was set up, why 30 cm, what will be the product version of it, and why not attach to the mechanical setup? These are important and there are fundamental consequences of such an arrangement. It is also not true that the surrounding environment will have little effect on the measurement.

Depending on the location of the microphone in different environments the received signals will have different multipath channel effect that need to be overcome. This can be done by initialization of the device with channel sounding test and tunable equalizer setting to cancel all the ghost images first. This initial training process needs to be done before starting the measurement like all the cell phones do to combat multipath issues and operate in a city environment. If this is not done, the ghost copies of source sound generated from reflections will overlap with the original copy and distort the signals. The authors cannot assume it will always be lucky and the private home environment will not create significant reflections like in a small bathroom with close solid walls surrounding the device.

The authors shall not assume the LSTM training will take care of the multipath issue since it will not and furthermore there are questions about the usage of LSTM itself as described in the following.

2. About using LSTM:

a. It is not clear why the author chooses LSTM and why RNN is not sufficient to obtain measurement results. How long the memory needs to be kept, which disqualified the usage of RNN and what memories need to be forgotten were not described. The reasons for not using RNN or even just conventional signal processing techniques were not explained and justified.

b. There was no clear description how Fig. 4 was obtained. The classification sound characteristics (a)-(c) are the ground truth or learned from already specified patients under supervised learning? In either case, conventional signal processing like cross-correlation calculations, ...etc. shall be sufficient to separate and classify the different groups. Again why LSTM needs to be employed. A comparison is needed to list the advantages of using LSTM.

c. The usage of the device with LSTM shall have training phase and deployment phase. It is not clear whether training will be done at the moment the products go out to the customers or further adjustment can be done to fit to the user environment. Will the work allow further on the fly training?

d. It is also unclear about how Table2 was obtained. The 27 subjects’ measurement results were used to train the LSTM and the trained setting was used to evaluate the 27 subjects? How were the classification rates obtained?

Author Response

(The authors gave the same response as above.)

Round 2

Reviewer 2 Report

Authors have address comments and made best effort to provide missing explanations and improve clarity and reproducibility of their work. Methodology and data preprocessing are sufficiently described. Reported results clearly show validation outcome of the proposed model. Readers will be interested in finding out new and promising model to record and analyse flow patterns and volumes in home settings. 

Author Response

Thank you for inviting us to submit a minor revised draft of our manuscript. We appreciate the time and effort by the reviewer to provide insightful feedback for clear presentation of our work. We agree with your suggestion. This study proposed a new method for predicting the flow velocity using the sound characteristics of urinary sounds that can replace the traditional flowmetry method and aims to apply it to hospitals. In general, hospitals conduct experiments in sound absorption-treated spaces to ensure patient privacy, allowing them to conduct the research without considering multipath channel effects. However, to realize in a variety of spaces other than hospital, it requires to solve the problem of signal distortion caused by the sound reflection. It is expected to conduct research related to home settings in the future. 

Reviewer 3 Report

The authors have made good progress to improve the manuscript.

But three places still need to be improved. 

  1. In Fig. 4 caption and/or the text,  the authors shall clearly add in what explained in the response writeup "the result of classifying patients' health conditions based on doctors' clinical experience through uroflowmetry results". The information hasn't been added into the MS.
  2. Need to add a short paragraph " compared with conventional methods what are the advantages of using LSTM. (this has been asked in first review but was not addressed)
  3. The response to comment-d is more confusing. So, the authors have more sets of data besides the original data sets. PLEASE describe data sets clearly and which were used for training and which were used for evaluation to obtain table 2. 

Author Response

Thank you for inviting us to submit a minor revised draft of our manuscript. We appreciate the time and effort by the reviewer to provide insightful feedback for clear presentation of our work. 
